# The Application of Prussian Blue Nanoparticles in Tumor Diagnosis and Treatment

**DOI:** 10.3390/s20236905

**Published:** 2020-12-03

**Authors:** Xiaoran Gao, Qiaowen Wang, Cui Cheng, Shujin Lin, Ting Lin, Chun Liu, Xiao Han

**Affiliations:** College of Biological Science and Engineering, Fuzhou University, Fuzhou 350108, China; N185720009@fzu.edu.cn (X.G.); wendyeveryday@163.com (Q.W.); linshujin32@163.com (S.L.); lintingtt33@163.com (T.L.); ibptlc@fzu.edu.cn (C.L.); hanxiao@fzu.edu.cn (X.H.)

**Keywords:** Prussian blue nanoparticles, immunosensors, imaging, drug delivery, imaging-guided therapy

## Abstract

Prussian blue nanoparticles (PBNPs) have attracted increasing research interest in immunosensors, bioimaging, drug delivery, and application as therapeutic agents due to their large internal pore volume, tunable size, easy synthesis and surface modification, good thermal stability, and favorable biocompatibility. This review first outlines the effect of tumor markers using PBNPs-based immunosensors which have a sandwich-type architecture and competitive-type structure. Metal ion doped PBNPs which were used as T_1_-weight magnetic resonance and photoacoustic imaging agents to improve image quality and surface modified PBNPs which were used as drug carriers to decrease side effects via passive or active targeting to tumor sites are also summarized. Moreover, the PBNPs with high photothermal efficiency and excellent catalase-like activity were promising for photothermal therapy and O_2_ self-supplied photodynamic therapy of tumors. Hence, PBNPs-based multimodal imaging-guided combinational tumor therapies (such as chemo, photothermal, and photodynamic therapies) were finally reviewed. This review aims to inspire broad interest in the rational design and application of PBNPs for detecting and treating tumors in clinical research.

## 1. Introduction

To date, cancer is still one of the most prevalent and deadliest diseases globally, with more than 200 different types of cancer that can result in more than 60 organ dysfunctions [1]. Increased mortality rates are mainly associated with the metastasis of solid tumors produced by large masses of tissue in the majority of cancer-related cases. In general, the limitation of early tumor diagnosis makes it less likely that tumors can be detected in the optimal treatment window. Once the most treatable stage passes, the tumor is likely to spread uncontrolled through the body by damaging the majority of healthy tissue, and this often leads to death [2,3]. In view of these issues, developing diagnostic techniques can be helpful for the early detection of tumor metastasis and their timely treatment using therapeutic agents that provide new possibilities for the efficient treatment of tumors.

Importantly, the level of tumor markers in human serum is much higher for patients with certain types of tumors. This can be a specific indication of the presence of tumors for early identification in clinical applications [4,5]. In the early stages of the disease, only trace amounts of biomarkers exist in body fluids. Hence, the development of sensitive and accurate detection techniques is of importance. Electrochemical immunosensors have been rapidly developed as alternatives to traditional immunoassay methods for determination of tumor markers; this has been achieved by combining the specific interactions between the antigen and antibodies with a convenient device structure. Due to their high specificity, miniaturization, low cost, real-time detection automation, and the capacity for multi-target analyses compared with traditional immunoassay methods, the resulting device can meet the requirements of point-of-care testing (POCT) and provide sensitive and convenient molecular diagnosis for underserved populations or community health-care systems [6,7,8]. Prussian blue (PB) is an iron centered compound (Fe_4_[Fe(CN)_6_]_3_· × H_2_O, where iron atoms have different oxidation states (Fe^2+^/Fe^3+^) and x is the number of water molecules) and is one of the oldest coordination compounds reported in the literature [9,10]. Since the accidental discovery of PB in 1704 by Diesbach, it took 274 years for its application as an electroactive film [11,12]. The development of PB was reported by Chu [13] in detail. Interstitial modifications and substitution of PB can generate a series of new coordination compounds, which are referred to as PB analogs (PBAs) [14]. PB and PBA, as a peroxidase analog, can provide a sensitive electrochemical response by catalyzing the reduction of hydrogen peroxide (H_2_O_2_) at lower dosages and supporting the integration of the oxidase enzymes while decreasing other electrochemical interferents [15,16,17]. A tuned core-shell, hollow, or substrate-supported PB/PBA framework enables electrolyte diffusion to enhance electrochemical performances [18]. Furthermore, the application of different nanomaterials such as graphene, carbon nanotubes, quantum dots, magnetic particles, and gold nanoparticles deposited on PB/PBA offer new avenues to increase the sensitivity [19,20]. In previous studies, PB/PBA-based biosensors were considered to be one of the most efficient electron-transfer mediators for applications including the determination of tumor markers, antihypertensive drugs, glucose, lactate, ascorbic acid, DNA, ions, and pH [21,22,23,24,25,26,27,28]. Thus, several strategies for signal amplification have been used to enhance the electrochemical response. Furthermore, strategies such as the application of different nanomaterials have been applied to improve the sensitivity, which is highly desired for detecting ultra-low amounts of target tumor markers.

With the continued development of nanoscience and nanotechnology, theranostics nanoplatforms have also been engineered for precision treatment of tumors by combining therapy with suitable imaging modality. This can be used to evaluate and optimize the therapeutic effect in real time via visual monitoring of the therapeutic procedure [29,30]. PB nanoparticles (PBNPs) have been reported in numerous applications, including for early tumor diagnostics via medical imaging such as T_1_-weight magnetic resonance imaging (MRI), photoacoustic imaging (PAI), and multi-model imaging [31]. It is reported that PBNPs were demonstrated as an effective T_1_-weighted cellular MRI contrast agent in 2010 [32], but it is approximately an order of magnitude lower than clinical T_1_-weighted MRICAs due to their poor ordinary longitudinal relaxivity. So, promoted their diagnostic capability by achieving a larger r_1_ with a smaller amount of injected PBNPs metallic ions has become a topic of intense research interest in recent years [33,34,35,36,37,38]. The PBNPs with a blue appearance can cause transient thermoelastic expansion via strong NIR absorption and provide ultrasonic signals for PAI [39], and modified PBNPs have better photothermal stability and photoacoustic signal-to-noise ratio [40]. PAI technique via PBNPs as a CA to label tumor or lable mesenchymal stem cells has been broadly exploited for noninvasive imaging and timely monitoring the recovery process [41,42,43], because it has high spatial resolution, strong contrast, noninvasiveness, real-time imaging, low cost, and the ability for both endogenous and exogenous imaging [44]. Hollow or mesoporous PBNPs have been usefully employed as nanocarriers to deliver theranostic agents because of their large intrinsic internal pore volume, tunable size, and simple synthesis [45]. Additionally, surface modification enables the PBNPs to effectively accumulate drugs at tumor sites via passive or active targeting, which overcomes the weaknesses of conventional drugs, including systemic toxicity, short circulation time, inefficient target specificity, and poor solubility [46,47]. In addition, PBNPs, which are an excellent candidate for photothermal therapy (PTT), have high optical absorption properties in the NIR region between 650–900 nm because of intervalence charge transfer between Fe^2+^ and Fe^3+^ in PB. This enables tumor eradication via cytotoxic heat generated from the PBNPs [48,49]. PBNPs can catalyze H_2_O_2_ into O_2_ in the existence of H_2_O_2_ to realize O_2_ self-supplied mode within solid tumors, and then improve the hypoxia of tumors and provide conditions for photodynamic therapy (PDT). It can also control the H_2_O_2_ decomposition rate via local temperature variation of PBNPs under NIR laser irradiation [50]. More importantly, several reports demonstrated that PBNPs have good biocompatibility and nontoxicity because of the tight bond between the cyano groups and iron [51,52]. Therefore, PBNPs-based composites might serve as a promising theranostics nanoplatform and overcome limitations such as a lack of accuracy, sensitivity, and real-time diagnosis. Additionally, therapy limitations such as resistance, inefficiency, and control in the biomedical field could potentially be overcome.

In this review, we briefly discuss recent advancements in the biomedical field involving the properties of PB-based nanoobjects as electron-transfer mediators for immunosensing, nanoprobes for imaging, therapeutic agents, and as nanocarriers for drug delivery (Figure 1). The application of PBNPs in the biomedical field in multimodal diagnostic techniques (biosensing or imaging) and therapy (chemo, PTT, PDT) of tumors have been summarized. In addition, in the tumor theranostic section, we mainly focused on multimodal imaging-guidance using combinational strategies for tumor therapy rather than individual ones. This review hopes to inspire broader consideration for the rational design of systems for the detection and treatment of tumors in clinical research.

## 2. PBNPs for Tumor Diagnosis in Electrochemical Immunosensors

For the screening and diagnosis of tumors, intensive research has gone into choosing specific and sensitive methods to improve the determination of tumor markers. The concentration of these biomarkers can be used to determine the stages of the tumor [53,54]. The enzyme-linked-immunosorbent assay (ELISA) is a conventional commercially available test for clinical diagnoses. However, some restrictions exist related to the relatively weak sensitivity, the detection levels of the protein in the advanced disease stage, long analysis time, high costs, and difficulty in application in the absence of a classical diagnostic laboratory. Electrochemical immunosensors, as a type of biosensor, monitor the content of analytes in a sample via recognition between the antibody and antigen (target analyte), and it has attracted increasing interest in tumor diagnosis because of its low instrumentation cost, simplicity, compactness, and automated testing capabilities [55]. Electrochemically depositing PB layers onto platinum was first reported by Neff in 1978 [11]. A few years later, Itaya et al. developed a versatile method for the deposition of a thin layer of PB on different electrode materials (SnO_2_, glassy carbon, gold, platinum) and demonstrated high stability after 105 cycles [56,57]. They also reported that PB had the capability to reduce O_2_ and H_2_O_2_ in the reduced form and oxidize H_2_O_2_ in the oxidized form; additionally, it could function as an electron transfer mediator because of the generation of two-electron transfer channels via the redox reactions in PB (high-spin Fe^3+/2+^ and low-spin [Fe(CN)_6_]^3−/4−^). The peculiar structure of PB allowed the diffusion of O_2_ and H_2_O_2_ through the crystal [58]. Then, Karyakin et al. reported a first-generation amperometric glucose biosensor that used PB as an electrocatalyst for H_2_O_2_ in 1994 [59]. Therefore, PB, as a typical electron transfer mediator and a H_2_O_2_ catalyst, has been widely used in electrochemical biosensors [15,60].

In particular, we focus on electrochemical immunosensors for the detection of multiplex tumor biomarkers in human serum (e.g., carcinoembryonic antigen (CEA), alpha-fetoprotein (AFP), carcinoma antigen 125 (CA125), cancer antigen 15-3 (CA 15-3), apolipoprotein A1 (ApoA1), and cytokines) based on the use of PB deposited on graphene, carbon nanotubes, quantum dots, magnetic particles, and gold nanoparticles as novel strategies for improved early tumor diagnostics. Generally, the most popular types of immunosensors are the sandwich and competitive types. (Figure 2) For the sandwich type, the analyte is first captured by the corresponding antibody and is then detected using another labeled antibody [8]. Dai et al. reported that PB modified hydroxyapatite was used as a carrier of the tracer (horse radish peroxidase (HRP)) and secondary anti-AFP antibodies (Ab_2_), and thionine (TH), and graphene sheets (GS) were used to capture the anti-AFP antibody (Ab_1_). This concept showed an increase in the value of current with increasing AFP concentration in the range of 0.02 to 8 ng/mL, with a limit of detection (LODs) of 9 pg/mL [61]. Generally, the average concentration of AFP in healthy adults is as low as 3.4 ng/mL [62]. This immunosensor prepared by Dai et al. has the advantages of high sensitivity and selectivity, and has been used applied to the analysis of AFP with satisfactory results. An increased concentration of tumor necrosis factor-α (TNF-α) was also associated with the tumor, while the amount of TNF-α in the human body was very low (10–30 pg/mL) [63]. It has been reported that PB modified ceria nanoparticles were used as an electrochemical label for the detection of TNF-α because of its high sensitivity for H_2_O_2_ detection. The proposed sandwich-format immunosensor had a wide linear range (0.005–5 ng/mL) and a LOD of 2 pg/mL for TNF-α, which exceeded the LODs of the ELISA technique in the human body (Figure 2a) [21]. So, it is important to investigate the structural design of different nanoparticle-based catalysts, which has been used as new supporting matrix for PBNPs, to enhance the catalytic activity and obtain synergistic effect. Nevertheless, most tumors have more than a single marker associated with their presence. Simultaneous determination of multiple biomarkers can enhance the diagnostic specificity between the biomarkers and tumor [64,65]. Lai et al. designed a screen-printed carbon electrode arrays (SPCEs) for ultrasensitive multiplexed measurement of tumor markers using PB as a mediator to catalyze the H_2_O_2_ produced by HRP with CEA and AFP as analytes. The LODs of AFP and CEA were 1.4 and 2.2 pg/mL, respectively [66]. HRP immobilized on PBNPs was a good method to enhance the enzymatically catalytical signal, and it could eliminate the electrochemical cross talk by the use of electrode arrays and single-enzyme label. However, the HRP label is environmentally unstable, requiring a nitrogen atmosphere and deoxygenation process, and the electrode arrays are not reusable resulting in a high-cost system [67,68]. Chen et al. simultaneously obtained LODs of 0.05 ng/mL for AFP and 0.1 ng/mL for CEA using a sandwich-type immunosensor without enzymatic reaction, which fully satisfied the requirements for clinical diagnosis with threshold values of 10 ng/mL^−1^ for AFP and of 3 ng/mL^−1^ for CEA in normal human serum; though, the LOD values were lower than for other studies with enzymatic reactions [69]. Therefore, if different antibodies were immobilized on the electrode at the same time, the low detection limit was got without enzymatic reaction. It could also help to avoid the use of the throwaway electrode arrays, which made the detection cheaper.

Another immunosensor format is the competitive type. Here, the signal of the antigen in a sample can be detected from the labeled antigen via competition between the labeled antigen and antigen in the sample, which reacts with the immobilized antibody. The detected electrochemical signal decreases with increasing analyte concentration [8,71,72]. Li et al. developed a competition-format immunosensor with low fouling for CA 15-3 detection using PEG as an antifouling polymer to decrease the adsorption of the nonspecific protein. Moreover, they considered Fe_3_O_4_ nanoparticles could release Fe^3+^, which triggered the growth of PB on the sensing interface and further generated and amplified the signals detected by the immunosensors [73]. Pure CA 15-3 and CA 15-3 functionalized Fe_3_O_4_ (Fe_3_O_4_@Ag) were thus used to competitively bind to the immobilized CA 15-3 antibody. Then, the signals for sensing were generated by PB due to electrochemical conversion of Fe_3_O_4_. The results showed that the amount of Fe_3_O_4_@Ag bonded to the electrode surface decreased with increasing concentration of CA 15-3, which is because of the competitive immunoreaction that causes a reduction in the detectable signals. A linear relationship between the current and the logarithm of the CA 15-3 concentration was obtained in the range of 10 to 1000 μU/mL, with a LOD of 5.2 μU/mL (Figure 2b) [70]. It is seen that the combination of the traditional electrode and a magnetic electrode could be a new strategy to generate and amplify the detecting signals for sensing. In general, compared with sandwich-type immunosensors, the competition-type immunosensor is less complex and has a lower incubation time, but it has a smaller linear range. Both types of immunosensor can be easily extended to detect various biomarkers.

The better clinical utility related to early detection has highlighted the need for sensitive tumor markers. The improved recognition of PB electrochemical behavior has been demonstrated, and it is expected that PBNPs-based immunosensors can be promising for screening, diagnosis, and prognosis of different tumors. This can lead to the sensitive and early detection of tumor markers for cancer prevention in high-risk individuals or to effectively eradicate tumors at an early point with a low tumor load.

## 3. PBNPs for Tumor Diagnosis in Biological Imaging

This section may be divided by subheadings. It should provide a concise and precise description of the experimental results, their interpretation as well as the experimental conclusions that can be drawn.

### 3.1. Magnetic Resonance Imaging

Magnetic resonance imaging (MRI) is a non-invasive imaging tool that has several advantages, including a high spatial resolution, high signal penetration depth, non-ionizing radiation, excellent soft tissue contrast, and wide clinical applicability. In practical clinical trials, CAs are used to further enhance the MRI sensitivity and obtain improved image quality by changing the relaxation time to increasing the relaxation rate around the agents. In general, MRI CAs are primarily T_1_-weighted agents with a positive image effect (bright spots) and T_2_-weighted agents with a negative image effect (dark spots) [16]. T_1_ contrast agents are conducive to bright spots because they increase the signal intensity in T_1_-weighted images via a shortened T_1_. T_2_ contrast agents enable dark spots because the T_2_ contrast effect reduces the signal intensity in the T_2_-weighted images [74]. Because it is difficult to recognize the dark region from the background, the T_2_-weighted imaging is not satisfactory [75]. In 2010, Shokouhimehr et al. first demonstrated that PBNPs with single-crystal-like feature were an effective T_1_-weighted cellular MRI contrast agent [32]. A PBNP is a mixed-valence iron hexacyanoferrate with the approximate formula A_4x_Fe^III^_4_[Fe^II^(CN)_6_]_3+x_·nH_2_O (A = Li^+^, Na^+^, K^+^, Rb^+^, Cs^+^, NH^4+^, Tl^+^; 0 ≤ x ≤ 1, n = 14–16; abbreviated as PBNPs). The Fe^3+^ centers with a high-spin (S = 5/2) have the ability to coordinate with water protons, and thus offer relaxivity. However, PBNPs have a poor capacity for MRI diagnosis and ordinary longitudinal relaxivity, which is approximately an order of magnitude lower than clinical T_1_-weighted MRICAs [33]. Increasing the longitudinal relaxation rate of PBNPs to promote their diagnostic capability has become a topic of intense research interest in recent years.

In practical clinical trials, CAs are used to further enhance the sensitivity of the MRI image quality by increasing the relaxation rate around the agents. For T_1_ CAs, the R_1_ relaxation rate is defined by [76]:R_1_ = 1/T_1_ = (1/T_1_)_0_ + r_1_ C,(1)
where R_1_ is the relaxation rate of the aqueous solution, T_1_ is the relaxation time in the absence of the contrast agent, C is the concentration of the contrast agent (mM), and r_1_ is its relaxivity (s^−1^ mM^−1^). According to Equation (1), the R_1_ relaxivity is inversely proportional to the T_1_ and proportional to r_1_ when C is a constant. The C of a CA is inversely proportional to its r_1_ when R_1_ is a constant. Consequently, for a constant CA concentration, shortening T_1_ is a way to increase r_1_. For CAs that satisfy the diagnostic requirements, designing CAs with a large r_1_ is one way to reduce its injection dose in clinical trials. Achieving a larger r_1_ with a smaller amount of injected PBNPs metallic ions is important, as this would reduce possible side effects compared with conventional CAs in biomedical imaging. This is based on the premise that the PBNPs remain fully stable under physiological conditions. Hence, it is necessary to carry out systematic toxicity tests on the internalization of the particles in the cells and determine the long-term stability of the PBNPs. Li et al. employed the Glypican-3 antibody to modify PBNPs for targeted MR imaging [33]. Subsequently, their group further demonstrated that PBNPs have acute toxicity in vivo. However, by monitoring the dynamic changes of the biochemical and immunity indicators, their long-term toxicity was determined to be low after short-term exposure [51]. For biomedical imaging application of PBNPs, a good ability for cellular uptake is an important pre-requisite. Shokouhimehr et al. demonstrated the cellular uptake of the PBNPs using MDA MB-231 cells and HeLa incubated with PBNPs. They also confirmed the non-toxic nature of the PBNPs [32]. Perera et al. fabricated PEG-coated Gd@PBNPs with long-term stability for gastrointestinal tract MR imaging. They demonstrated that the increased positive MRI signal in the stomach remained stable for 60 min, and remained visible even after 90 min via abdominal injection [34].

Inspired by good performance of Mn^2+^, Gd^3+^, and Fe^3+^ for enhancement of T_1_-weighted MRI, an effective strategy for improving r_1_ value is to dope metal ions in the PB nanocrystal structure. Typically, the following four areas are important for enhancing the MRI sensitivity and obtaining a good image quality: dopant effects, size control, testing temperature, and shape regulation. (1) Dopant effects: Mn^2+^ has five unpaired electrons (J = 5/2, S = 5/2, L = 0), Gd^3+^ has seven unpaired electrons (J = 7/2, S = 7/2, L = 0), and Fe^3+^ has five unpaired electrons (J = 5/2, S = 5/2, L = 0), making Mn^2+^-based PBNPs, Gd^3+^-based PBNPs, and Fe^3+^-based PBNPs promising candidates as T_1_ CAs. Paul et al. reported that the value of r_1_ could increase to 15 mM^−1^s^−1^ via doping Mn^2+^ in PBNPs, and the r_1_ value of Mn^2+^-doped PB was 40 times that of 7 nm MnO nanoparticles and about twice that reported for the best 2.5 nm MnO nanoparticles [35]. Fétiveau et al. designed two series of ultrasmall PBANPs with different Gd^3+^ contents for imaging tumors. They obtained outstanding longitudinal relaxivities of 40 mM^−1^s^−1^ (at 1.4 T) per Gd^3+^ and also confirmed that Fe^3+^ ions also contribute to the overall relaxivity (r_1_ = 55 mM^−1^s^−1^) to some extent [36]. (2) Size control: for T_1_ CAs, the r_1_ value could be enhanced by increasing the number of bound water molecules (q) and increasing the rotational correlation time (tR) on the basis of the classical Solomon Bloembergen Morgan (SBM) theory [76]. In the small metal ion-doped PBNPs, the metal ions should be located at the surface, making these metal ions sites particularly accessible to water and this increases the number of bound water molecules. Ali et al. synthesized different sizes of Mn^2+^-doped PBNPs and revealed that the r_1_ value was equal to 2.58 and 5.30 mM^−1^s^−1^ for sizes of 73 nm and 61 nm, respectively [37]. (3) Testing temperature: according to SBM, the r_1_ value increased as the temperature decreased because of the increase in the rotational correlation time. The Paul group reported that the r_1_ of Mn^2+^-doped PBANPs and Gd^3+^-doped PBANPs increased when the temperature decreased from 37 to 5 °C [35,36]. (4) Choice of shape: hollow mesoporous PBNPs have been demonstrated as excellent drug carriers to load Mn^2+^ ions and be exposed to an aqueous environment, which can be triggered by the release of Mn^2+^ in the site of interest (e.g., tumor site). This makes it a smart T_1_-weighted MRI CA with an ultrahigh relaxivity for tumor diagnosis [38]. Table 1 summarizes the MR relaxivities based on PBNPs for this section.

### 3.2. Photoacoustic Imaging

The photoacoustic imaging (PAI) technique is a broadly applicable noninvasive imaging modality; transient acoustic waves are generated via the absorption of NIR in biological tissues, which can then be detected using a traditional ultrasound transducer [77,78,79,80]. PAI has numerous advantages including a high spatial resolution, strong contrast, noninvasiveness, real-time imaging, low cost, and the ability for both endogenous and exogenous imaging, which has attracted wide attention for applications involving the imaging of biological tissues [44]. However, the low laser absorption of the endogenous chromophore leads to a weak signal, which is a major obstacle for deep tissue imaging [81]. Choosing a good NIR-absorbing photoacoustic imaging contrast agent (PAICA) is a necessary condition to ensure the high-quality imaging of biological tissues.

The PBNPs with a blue appearance can cause transient thermoelastic expansion via strong NIR absorption and provide ultrasonic signals for PAI [39]. Several groups have demonstrated the feasibility of PBNPs as a PAICA. Liang et al. first demonstrated that PBNPs have efficient absorption of NIR at 765 nm, which is within the optical window for the deepest light penetration in tissues and can be a superior PAICA [82]. Zheng et al. revealed that PB provided a distinct photoacoustic signal in the tumor and the signals showed a time-dependent increase [83]. Xu et al. developed an “in situ modification” synthesis to obtain good physiological stability of PB for over 90 days using PVP and K_3_[Fe(CN)_6_]. They further reported that the PAI signal intensity of PB was brighter than the control group in tumor-bearing mice, and it exhibited an obvious concentration-dependence with injection of PB [41].

The good photothermal stability of PBNPs after NIR irradiation is vital for improving the photoacoustic signal-to-noise ratio. Dumani et al. synthesized PB nanocubes using superparamagnetic iron oxide nanoparticles (SPION) as precursors and verified their photothermal stability under pulsed laser irradiation. Unlike gold nanorods (degradation at 8.5 mJ·cm^−2^) and silica-coated gold nanorods (degradation at 13 mJ·cm^−2^), the PB nanocubes (PBNCs) displayed no degradation of the photoacoustic signal (up to 28 mJ·cm^−2^). Importantly, the photoacoustic signal from the injection of PBNPs was approximately five-fold higher than the control injection after 700 nm laser irradiation [40]. More importantly, during the course of clinical treatment, PBNPs have been used as a stem cell tracer to monitor the dynamic process of migration and differentiation owing to cellular internalization via endocytosis. Stem cells have some ability for migration, differentiating into various functional cells, interaction with pathology, and self-renewal that are vital to understanding the status of disease recovery. Kubelick et al. demonstrated that PBNCs have ability to label stem cells and generate contrast for PAI. Their results indicated that PAI with PBNCs can provide real-time feedback on mesenchymal stem cells (MSCs) delivery and motivated an approach for stem cell tracking in the spinal cord [42]. The PAI technique via PBNPs as a CA to label bone MSCs has also been exploited for imaging traumatic brain injury and monitoring the recovery process [43].

## 4. PBNPs for Drug Delivery

An ideal drug delivery system is required to have low cytotoxicity, excellent stability and biocompatibility, large loading capacity, and responsive release [84]. Hollow or mesoporous PBNPs with large intrinsic internal pore volumes, tunable size, and easy synthesis have been successfully employed as carriers to deliver drugs [16]. However, bare hollow PBNPs (HMPBNPs) would lead to low therapeutic efficacy and high toxicity because of drug leakage, uncontrolled drug delivery, and a low immune evading ability. Recently, HMPBNPs with surface modification have been proposed as a new class of drug delivery system for chemotherapy. Table 2 summarizes some PBNPs for drug delivery.

For specific tumor chemotherapy, a multi-responsive smart drug delivery system is expected to enhance antitumor treatment and reduce toxicity via prolonging the systemic circulation time, decreasing the identification of monocytes and macrophages, targeted accumulation at the tumor site, and controlled drug release. Recently, PBNPs combined with natural biomaterials, such as red blood cell (RBC) membranes have drawn research attention to facilitate the accumulation of drugs at tumor sites via passive targeting and extension of the circulation time. Liu et al. first constructed a nanodrug system using RBC membrane-coated HMPBNPs to extend the blood circulation time to 10 h and increase the immune evasion ability to more than 60% for tumor targeting and long circulation therapy. Additionally, active targeting drug delivery systems are most widely used for tumor therapy to improve the selectivity and specificity via chemical synthetic strategies. Hyaluronic acid (HA) was introduced to bind to the CD44 receptor of breast cancer cells and increase the tumor-targeted accumulation of the drug. Based on the effective accumulation of nanocomposites at tumor sites, anti-tumor drug gamabufotalin (CS-6) was loaded by taking advantage of the high surface volume of HMPBNPs for chemotherapy [46].

In addition, controlled release of PBNPs-based nanocomposites for tumor therapy can be triggered via chemical (pH or redox) or physical (heat or light) stimuli [85]. Doxorubicin (DOX), as a chemotherapy drug with excellent anti-tumor and pH-responsive effects, is used for the assessment of the accumulation at tumor sites and the release pattern in the drug delivery system. Xiao et al. used the erythrocyte membrane (EM) and folate (FA) to modify spheric and cubic hollow mesopores PBNPs (SCPBNPs) and introduce DOX. The results demonstrated the potential in the SCPBNPs@DOX@EM@FA treatment group for reducing macrophage phagocytosis and immune response, extending the blood circulation time (3.3-fold) and the half-life (2.5-fold), and improving targeting (2.5-fold) compared with the bare SCPBNPs treatment group for tumor therapy [47]. Based on the pH-responsive effect of DOX, a bimetallic NiCo-PB analog (NiCo-PBA) doped Tb^3+^ was embedded in a PEGMA layer and then anchored with the AS1411 aptamer and DOX; it demonstrated a pH-responsive capability, controlled drug release within cancer cells, and enhanced tumor-targeted delivery of DOX [86]. In another study, Li et al. developed DOX-encapsulated PBNPs coupled pentacarbonyl iron (Fe(CO)_5_) as a CO producer and decorated it with hydrophilic polymers via a layer-by-layer technique to enhance its aqueous stability. They further demonstrated that Fe(CO)_5_ was cleaved at 42 °C to release CO, which caused mitochondrial collapse, inhibition of APT-dependent drug efflux, and enhanced drug accumulation in the cancer cells to overcome multidrug resistance and improve the anti-cancer effect under the NIR light irradiation (808 nm, 0.5 W/cm^2^) [87].

## 5. PBNPs for Tumor Photothermal and Photodynamic Therapy

This section may be divided by subheadings. It should provide a concise and precise description of the experimental results, their interpretation as well as the experimental conclusions that can be drawn. In clinical tumor therapy, photothermal therapy (PTT) activated using NIR light irradiation has attracted significant attention because of its advantages, such as deep tumor penetration, non-invasiveness, and precise remote control [88]. PBNPs have high optical absorption properties in the NIR region from 650 to 900 nm because of intervalence charge transfer between Fe^2+^ and Fe^3+^ in the PB structure, which can realize tumor eradication via the cytotoxic heat generated from PBNPs [89]. Fu et al. first reported PBNPs as new PPTCAs to treat tumors under NIR irradiation (at 808 nm). They demonstrated that PBNPs have high molar extinction coefficients (1.09 × 10^9^ m^−1^cm^−1^) that are of the same order of magnitude as conventional Au nanorods (5.24 × 10^9^ m^−1^cm^−1^), which can decrease tumor cell viability to 10% after treatment using NIR irradiation [90]. In addition, the good photostability, biocompatibility, and dispersibility of the PBNPs-based nanocomposites under physiological conditions have promoted their development as PTTCAs for tumor treatment.

As a PTTCA, the high photothermal conversion efficiency of PBNPs is vital for increasing tumor treatment efficiency, and this is determined by their chemical composition, size and irradiation time, and the power and photon excitation mode. Changing the chemical composition of PBNPs by doping with metal ions and coating with polyethylene glycol could promote the photothermal conversion efficiency and increase the stability under physiological conditions. Coating PBNPs on NaNdF_4_ nanoparticles could increase photothermal conversion efficiency, currently at 60.8% under 808 nm laser irradiation, via the generation of new cross relaxation pathways between the Nd^3+^ ions and PBNPs compared with stand-alone NdNPs and PBNPs [89,91,92]. Doping lanthanide ions into PBNPs was achieved via a simple method, and it was used to improve the photothermal conversion efficiency (PBNPs-ytterbium, 55%) by changing the electron density of the cyanide bonds in PBNPs and producing luminescence for optical super resolution imaging in deeper tissue (PBNPs-erbium) [89,93]. In another study, Shou et al. demonstrated that the particle size of the PBNPs could be controlled by Zn^2+^ doping (PBZnNPs (n%)) and the photothermal conversion efficiency decreased as the particle size increased and as the irradiation time and power decreased [94]. Additionally, it has been demonstrated that two photon excitation (TPE) can enable improved penetration depth and laser focus compared with the common single photon excitation (SPE) mode, which can also influence the PTT efficiency. For this reason, Ali et al. investigated the efficiency of combined PTT and chemotherapy to eradicate tumor cells and clearly reported a significant advantage of TPE laser irradiation over SPE irradiation under the same experimental conditions treated with nanoparticles (Figure 3). Data showed that, in the case of pulsed TPE at 808 nm, 91 % of cell death was found with 1@DOX 24 h after irradiation (Figure 3a), but the cells treated with 1@DOX only showed 35 % of cell death after 24 h of irradiation (808 nm, 2.5 W/cm^−2^) (Figure 3b)[95].

Photodynamic therapy (PDT) that is minimally invasive and simple has also been widely studied for clinical treatment of localized and superficial tumors by generating excess reactive oxygen species (ROS) to induce cell death. The characteristics of hypoxia in solid tumors result in invasion and metastasis of tumor cells and greatly limit the anti-tumor effect of PDT treatment [96]. It has been reported that high H_2_O_2_ levels in the tumor microenvironment are an appropriate source for O_2_ production [97]. PBNPs were found to possess catalase-like activity via the Fenton reaction, which can catalyze H_2_O_2_ into O_2_ [98,99]. The generated O_2_ can effectively support the PDT of tumors and reduce tumor growth and metastasis by decreasing the tumor hypoxia situation. Thus, PBNPs have been extensively used as catalysts to improve the tumor treatment efficacy in PDT. Zhang et al. demonstrated that HA with a macrophage transformation property (pro-tumor M_2_ to anti-tumor M_1_) modified HMPBNPs (HA-PB) can catalyze H_2_O_2_ for O_2_ self-supplied PDT and inhibit tumor proliferation. Subsequently, the photosensitizer indocyanine green (ICG) was loaded into HA-PB to realize the production of a singlet oxygen (^1^O_2_) under NIR irradiation with a 808 nm laser for tumor PDT [50]. Moreover, PBNPs are also expected to control H_2_O_2_ decomposition within the tumor by local temperature variation of the PBNPs under NIR laser irradiation to improve the therapeutic effect via the combination of PDT and PTT. Wang et al. realized the photothermally controlled improvement of hypoxic condition in cancer cells and tumor tissues using PB@SiO_2_-PEG-ZnPc (PSPZP) NCs. Figure 4a showed the synthetic process of PSPZP NCs with enhanced photo-induced O_2_ and ^1^O_2_ generation. As shown in Figure 4b, abundant O_2_ bubbles were formatted immediately once the PSPZP NCs were added into H_2_O_2_ solution, and the catalytic reaction rate showed an obvious trend of growth with the increase of temperature from 25 °C to 43 °C. Further, they measured the generation of O_2_ (Figure 4c), and the data showed that the generated O_2_ by PBNPs under 43 °C was 1.95 times that under 25 °C with 671 nm laser irradiation. Since improved H_2_O_2_ decomposition at higher temperature would increase PDT efficiency of PSP NCs by accelerating O_2_ supply, they observed lower cell viability with the increase of temperature after irradiation (Figure 4d), More importantly, the constructed nanoplatform could realize significant tumor inhibition in mice compared with PDT or PTT treatment (Figure 4e) [100]. In view of these facts, designing a multi-responsive smart drug delivery system with a PBNPs-based nanocomposite using a combined PTT/PDT technique is of interest for overcoming the limitations of PDT, such as O_2_ dependence non-selectivity, and the high laser power requirements of PTT [101,102,103].

## 6. PBNPs for Tumor Imaging-Guided Therapy

Imaging-guided therapy is a combination of medical therapy and diagnostic imaging to monitor the accumulation of a therapeutic agent within the tumor or enhance the therapeutic effects while decreasing the side effects during tumor treatment. It has shown great potential for drug delivery and image-guided minimally invasive therapy. For this reason, PBNPs-based imaging-guided therapy could decrease severe side effects such as a strong light sensitivity, normal cells lesions, and drug resistance in patients. It could also enhance therapeutic efficiency via association of the image-guided PTT or PDT technique with chemotherapy and develop a personalized synergistic therapy with real-time imaging diagnosis and tumor treatment. Table 3 summarizes the recent imaging-guided therapy strategies developed based on PBNPs.

As mentioned before, PBNPs-based nanoparticles as theranostics probes have been extensively used for tumor treatment with a single imaging modality, such as MRI or PAI [37,39,94,112,113]. Nevertheless, obtaining complete information and synchronous imaging diagnosis within the tumor using single-modal imaging is still a great challenge. For example, PAI has the advantages of excellent sensitivity, high spatial resolution, and real-time imaging, but it has a limited tissue penetration depth because of optical scattering [114,115]. MRI is a noninvasive technique for early tumor diagnosis because of its high spatial resolution and deep tissue penetration, but it still has the issue of poor sensitivity [116]. Combining two or more imaging modalities can allow for accurate collection of complementary and synchronous information on the same region of pathological tissue [117]. Multifunctional PBNPs, as contrast agents for various imaging modalities, with a high photothermal conversion efficiency, catalase-like activity, and drug loading capacity, have been explored for their synergistic effects with combined multimodal imaging-guided therapy for tumor treatment. Cai et al. reported a new strategy for functionalizing PB to obtain the controllability of the maximum absorbance in the NIR of PB from 710 to 910 nmby incorporating Gd^3+^ ions into the lattice sites of PB, meeting the demands of good spatial resolution and sensitivity, deep tissue penetration, real-time PAI/MRI, and excellent PTT performance, Gd^3+^-doped PBNPs showed an ability for scavenging ROS and avoiding oxidative stress injury to normal cells, demonstrating good biosafety and completely eliminating the tumor [104]. In another work, Peng et al. successfully fabricated a PB/MnO_2_ nanosystem that had efficient O_2_ production to oxygenate deoxygenated hemoglobin within the tumor tissue and improve the diamagnetic T_2_ signal intensity, which can be used as T_2_ modal contrast agent and also as a T_1_/PAI for imaging-guided PTT and oxygen regulation of the tumor [105].

In addition to MRI, integrating photosensitizers (PSs) in PAI-guided therapy is a common strategy; these include plasmonic, ICG, and semiconductor nanomaterials into PBNPs to obtain multiplex phototherapy and new imaging modalities under NIR laser irradiation. Core/shell nanoparticles of Au@PBNPs were fabricated for X-ray computed tomography (CT)/PA imaging-guided PTT of tumors. The Au core served as a CA for CT imaging to provide high resolution and 3D images, and the external PB shell afforded PAI and PTT effects [106]. Moreover, AuNPs are excellent surface-enhanced raman scattering (SERS) substrates that can significantly enhance the signal-to-background ratio (SBR) of the incorporated PBNPs. For this reason, Zhu et al. further modified Au@PBNPs by grafting HA for enhanced accumulation at the tumor site. The constructed theranostic agent possessed an enhanced T_1_ and T_2_ signal, stable zero-background single SERS emission, and photothermal/photodynamic conversion under NIR light irradiation, allowing for the realization of SERS/MR imaging-guided PDT/PTT for targeted-tumor treatment [107]. In addition, Peng et al. designed carbon dot-decorated PBNPs with a stable green photoluminescent property, efficient photothermal conversion capacity, and excellent biocompatibility. It showed excellent theranostics efficiency for fluorescence (FL) imaging-guided PTT for tumor treatment [108]. Conjugating ICG onto the surfaces of the PBNPs also has potential for application as a nanotheranostic agent for T_1_-weighted MR and FL imaging-guided PDT/PTT of tumors [109].

PBNPs incorporating different types of ultrasound contrast agents (UCAs) that serve as theranostic agents for multi-modal imaging and tumor therapy have made great progress. Jia et al. prepared a nanotheranostic agent for ultrasound imaging-guided PTT of tumors based on perfluoropentane encapsulated HMPBNPs. The heat produced by the PBNPs could cause gasification of the perfluoropentane to generate bubbles under NIR laser irradiation, which could improve the sensitivity in the tissue of interest during ultrasound imaging (USI) [118]. However, USI with a poor spatial resolution and contrast generally limits the early diagnosis of tumors [119,120]. Zhang et al. constructed perfluorohexaneen capsulated HMPBNPs loaded DOX for PAI/USI dual-modal imaging-guided chemo-high-intensity focused ultrasound (HIFU) therapy. HIFU could trigger DOX release and raise the temperature rapidly by depositing acoustic energy in the focal volume, which promoted the treatment of deep solid tumors. Dual-modal imaging overcame the limitations of USI by combining sensitive optical contrast and a high ultrasonic resolution [110]. More importantly, PBNPs can catalyze the breakdown of H_2_O_2_ into oxygen bubbles in the tumor, which changed the acoustic impedance of the tissue to make it detectable by USI [99]. Zhang et al. prepared an integrated nanoplatform of Mn^2+^-doped iron oxychloride (FeOCl) nanorods coated with PBNPs, polydopamine (PDA), and black phosphorus quantum dots (BPQDs), which could serve as MRI/PAI/USI agents for tri-modal imaging-guided CDT/PTT/PDT of tumors. Thereinto, the PBNPs, FeOCl, and Mn^2+^ with catalase-like activity could provide O_2_ for O_2_ self-supplied PDT by catalyzing the H_2_O_2_ within solid tumors. FeOCl catalysts had a high yield of hydroxyl radicals (•OH) by decomposing the H_2_O_2_ in chemodynamic therapy (CDT), and BPQDs enabled the transformation of O_2_ into ^1^O_2_ to enhance PDT [111].

Therefore, these theranostic agents hold promise for achieving a prolonged systemic circulation time, targeted accumulation at tumor sites, remote controlled drug release, and noninvasive therapy of tumors, which could increase the tumor selectivity and reduce side-effects. Additionally, they can also be exploited to provide tumor results and guidance on the timing and quantity of tumor treatment because of their imaging capabilities.

## 7. Conclusions

This review provides a summary of strategies that use PBNPs as an electron transfer mediator and as a promising theranostic agent for several biomedical applications involving tumors. We focused on the feasibility of using PBNPs as electrocatalysts to enhance the diagnostic specificity between the biomarkers and tumors via the simultaneous determination of multiple tumor markers in electrochemical immunosensors. We also reported on the different types of immunosensors using PB or in-situ generation of PB via the electrochemical conversion of Fe_3_O_4_ as a transduction platform. Compared with a commercial ELISA, PBNPs-based immunosensors are a powerful competitor for tumor diagnosis. To improve the therapeutic efficacy of tumors, we introduced the synergistic effects of imaging-guided multimodal therapies mediated by pure PBNPs as well as combinations with chemotherapeutic or chemodynamic agents. Furthermore, we discussed the influence of various factors on the MRI image quality, including dopant effects, size control, testing temperature and choice of shape. The PAI effect was influenced by photothermal stability and photoacoustic signal-to-noise ratio. We described the application of integrated photosensitizers and fluorocarbons into PBNPs to obtain multiplex phototherapy and new imaging modalities such as FL, SERS, CT, and US imaging. Additionally, we discussed hydrophilic polymers, metal ions, biological ligands, and cell membranes used for surface modification to prolong the systemic circulation time, improve targeted accumulation at tumor sites, and control the release of drugs on triggering by pH, redox, heat, or light stimuli. We summed up PBNPs-based photothermal treatment effect, which was determined by their chemical composition, size and irradiation time, and the power and photon excitation mode. PBNPs were extensively used as catalysts to improve the tumor treatment efficacy in PDT due to their ability of the generated O_2_ in H_2_O_2_ aqueous solution. Based on this review, the surveyed literature showed that PBNPs-based nanocomposites have excellent potential as an electron transfer mediator for sensors, as contrast agents for MRI and PAI, as drug carriers for anticarcinogen loading, as light absorbing agents for PTT and as a nanoenzymes for PDT to ensure high efficiency in tumor diagnosis and therapy. However, these studies are still in the early stages for the development of clinical PBNPs-based nanomedicines, and further progress is urgently required for practical clinical applications.

## Figures and Tables

**Figure 1 sensors-20-06905-f001:**
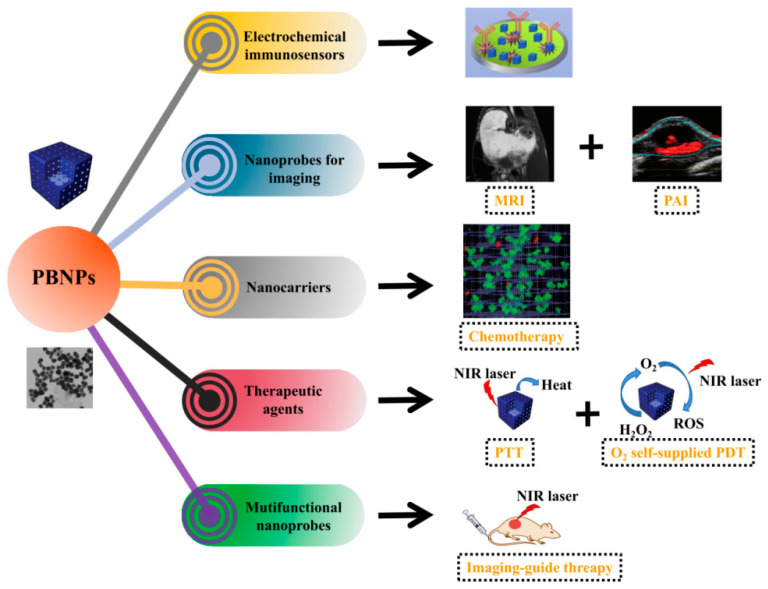
The schematic illustration of PBNPs in the applications of tumor diagnosis and therapies.

**Figure 2 sensors-20-06905-f002:**
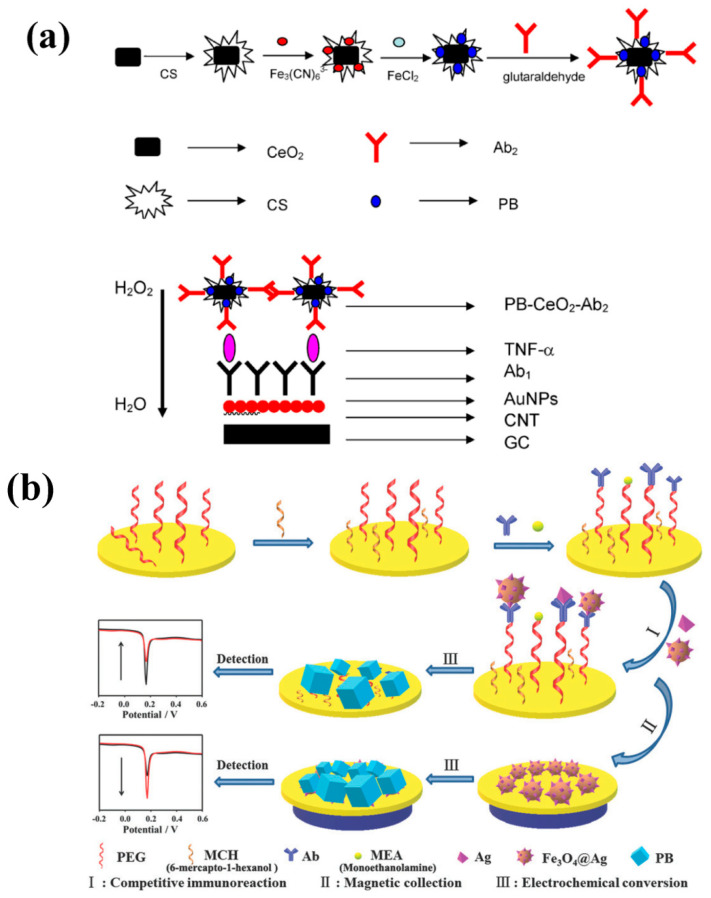
Schematic illustration of the electrochemical immunosensor fabrication process for sandwich format (**a**) and competitive format (**b**), adapted with permission from ref. [21,70].

**Figure 3 sensors-20-06905-f003:**
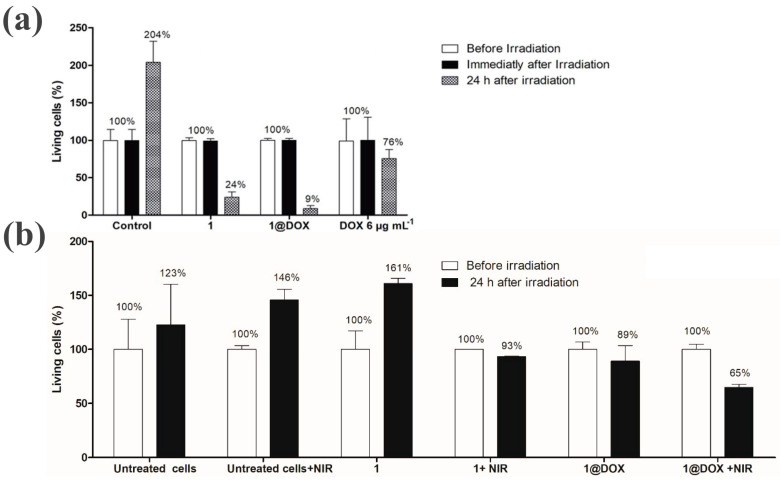
Cell counting (%) of living MDA-MB-231 cells treated with Mn^2+^-doped PBNPs@DOX (1@DOX) and Mn^2+^-doped PBNPs (1) at 50 mg/mL and free DOX before irradiation and 24 h after irradiation with a TPE laser at 808 nm (3.7 W, 5% of total laser power) for 10 min (**a**) and with SPE at 808 nm (2.5 W cm^−2^) for 30 min (**b**), adapted with permission from ref. [95].

**Figure 4 sensors-20-06905-f004:**
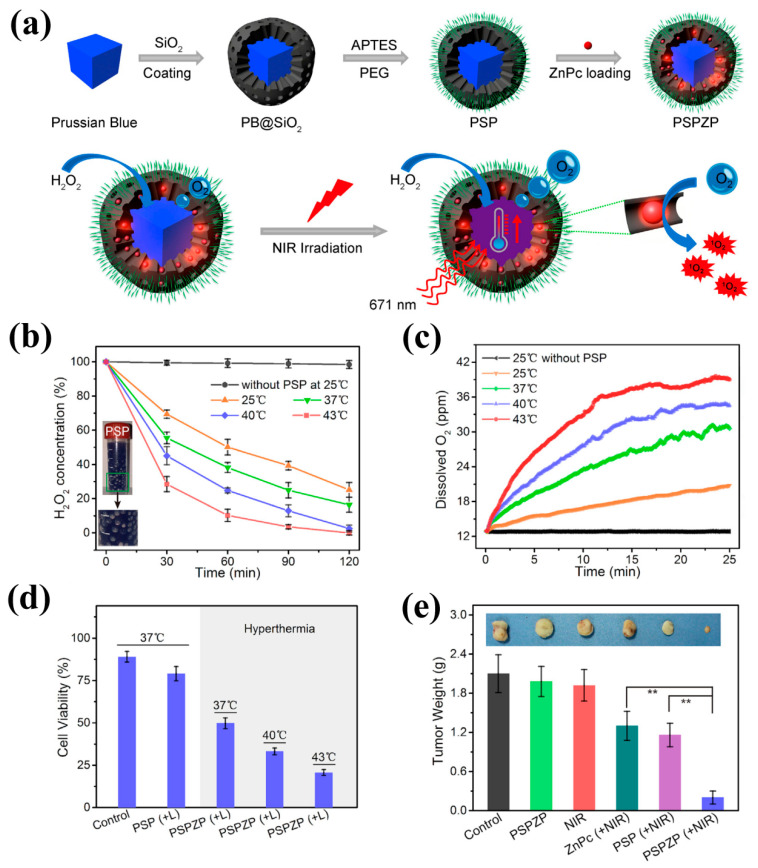
(**a**) Schematic of the synthetic procedure and photo-enhanced therapy of the PSP NCs; (**b**) Decomposition of H_2_O_2_ and (**c**) O_2_ generation treated with PSP NCs or not treated; (**d**) Relative viabilities of 4T1 cells incubated with PSP or PSPZP NCs; (**e**) Average weights of tumors collected from mice at the end of treatments, adapted with permission from ref. [100]. ** *p* < 0.01.

**Table 1 sensors-20-06905-t001:** Summary of MR relaxivities based on PBNPs.

NPs	Approximate Size (nm)	Surface Coating	Shape	r_1_ (mM^−1^s^−1^)	r_2_ (mM^−1^s^−1^)	Field (T)	T (°C)	Ref.
PBNPs	25	-	Cubes	0.2	1.22	1.5	-	[32]
				0.14	2.88	7		
AntiGPC3-PBNPs	21	Citrate	Cubes	0.14	11.73	9.4	20	[33]
Gd@PBNPs	24 ± 9	PEG	Cubes	16.4	20.9	1.4	37	[34]
Mn@PBNPs	5.5	Dextain	Sphere	12.9	-	3	5	[35]
				10	15		37	
Gd@PBNPs (GdFeFe)	3.9	Dextain	Sphere	81	-	1.4	5	[36]
				55	77		37	
Mn@PBNPs	71/63	-	Cubes	2.58/5.3	-	4.7	RT	[37]
Mn@HMPBNPs	290	PVP	Cubes	3.0	-	7.0	25	[38]

**Table 2 sensors-20-06905-t002:** Summary of PB-based nanoparticles for drug delivery.

Name	Drug Release	Size (nm)	Targeting	Anti-Tumor Drug	Loading Efficiency (%)	Circulation Time (h)	Ref.
HA@RBC@PB@CS-6	pH-/photo-responsive release	140	HA, RBC	CS-6	-	10	[46]
PB@DOX@EM@FA NPs	pH-/photo-responsive release	185	FA, EM	DOX	130.69	48	[47]
NiCo-PBA@Tb^3+^@PEGMA@AS1411@DOX	pH-responsive release	173	As1411	DOX	77.2	-	[86]
PAH@PAA@PEG@PB@CO@DOX	pH-/NIR light release	128	Passive targeting	DOX	14.5	-	[87]

**Table 3 sensors-20-06905-t003:** Summary of therapeutic PBNPs-based nanomaterials for imaging-guided tumor treatment.

Formulations	Treatment Approaches	Imaging	Targeting	Cell Lines	Laser Irradiation	Ref.
antiglypican-3-PBNPs	PTT	MR	Antiglypican-3	HepG2; HL-7702 cells	808 nm laser (2 W cm^−2^, 10 min)	[33]
PVP or dextran-coated Gd^3+^@PBNPs	PTT	MR; PA	-	CT26 cells	808 nm laser (1 W cm^−2^, 5 min)	[36]
Mn^2+^-doped PBNPs	PTT	MR	-	MDA-MB-231 cells	Two-photon light at 808 nm (3.7 W, 10 min)	[37]
PVP-coated HMPB-Mn	Chemothermal therapy	MR	-	4T1 cells	808 nm laser (1 W cm^−2^, 5 min)	[38]
Au@PB@Cu_2_O@BPQDs/PAHNCs.	PTT/PDT	MR; PA; FL	-	HeLa cells	650 nm laser (1.5 W cm^−2^, 5 min)	[83]
PBNPs	PTT	MR/PA	-	4T1 cells	808 nm laser (0.8 W cm^−2^, 5 min)	[41]
PB@DOX@EM@FANPs	Chemo-photothermal therapy	PA; FL	Folic acid	HeLa cells	808 nm laser (0.8 W cm^−2^, 5 min)	[47]
PBNPs@Fe(CO)_5_@DOX	Chemo-photothermal-photodynamic therapy	US	-	MCF-7/ADR cells	808 nm laser (0.5 W cm^−2^, 15 min)	[87]
NdNdF_4_@PBNPs	PTT	PA	-	HeLa cells	808 nm laser (0.6 W cm^−2^, 10 min)	[91]
Zn^2+^@PBNPs	PTT	MR	-	4T1 cells	808 nm laser (1 W cm^−2^, 5 min)	[94]
PB@SiO_2_-PEG-ZnPc	PDT; PTT	PET; PA	-	4T1 cells	671 nm laser (0.4 W cm^−2^, 5 min)	[100]
Gd^3+^@PBNPs	PTT	PA; MR	-	4T1 cells	808 nm laser (0.58 W cm^−2^, 10 min)	[104]
MnO_2_@PBNPs	PTT	PA; T1/T2 weighted MRI	-	MCF-7 cells	808 nm laser (2.5 W cm^−2^, 5 min)	[105]
Au@PB	PTT	CT; PA	-	HT-29 cells	808 nm laser (1.5 W cm^−2^, 10 min)	[106]
Au@PB-HA	PDT; PTT	MR; SERS	Hyaluronic acid	4T1 cells	808 nm laser (2 W cm^−2^, 10 min)	[107]
CD-decorated PBNP	PTT	FL	-	C6 cells	808 nm laser (0.8 W cm^−2^, 10 min)	[108]
PB-BSA-ICG	PDT; PTT	MR; FL	-	SCC7 cells	808 nm laser (1 W cm^−2^, 10 min)	[109]
HMPBs-DOX/PFH	Chemo-HIFU therapy	PA; US	-	VX2 cells	-	[110]
FeOCl@PB@PDA@BPQDs	CDT; PDT; PTT	PA; MR; US	-	4T1 cells	650 nm laser (1.5 W cm^−2^, 5 min)	[111]

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
