# Peer review of "The Application of Prussian Blue Nanoparticles in Tumor Diagnosis and Treatment"

_sensors, 2020, doi:10.3390/s20236905_

Round 1

Reviewer 1 Report

This review describes the use of prussian blue nanoparticles for the evolving landscape of cancer diagnostic and treatment. It highlights the different opportunities that such nanoparticles offers, specially in the electrochemical field, to be used as electron-transfer mediators for immunosensing. My recommendation is to accept this review as it is.  

Author Response

Thanks a lot to the reviewer’s positive comments.

Reviewer 2 Report

Here, authors reviewed the application of prussian blue nanoparticles (PBNPs) in tumor diagnosis and treatment such as electrochemical immunosensors, bioimaging, drug delivery and application as therapeutic agents. I agree that the author has consulted a large number of references to elaborate on the above aspects. However, I think that if the author can add more of his own opinions and analysis instead of listing references (line 120-160), it will better improve the quality of the paper. And I recommend the manuscript to be published in Sensors journal after major revision.

Author Response

Thanks a lot to the reviewer’s comments. We have added our own opinions and analysis. The modified parts are marked with blue color in the revised manuscript. That is,

“Generally, the average concentration of AFP in healthy adults is as low as 3.4 ng/mL [62]. This immunosensor prepared by Dai et al. has the advantages of high sensitivity and selectivity, and has been used applied to the analysis of AFP with satisfactory results.”

“So, it is important to investigate the structural design of different nanoparticles based catalysts, which has been used as new supporting matrix for PBNPs, to enhance the catalytic activity and obtain synergistic effect.”

 “HRP immobilized on PBNPs was a good method to enhance the enzymatically catalytical signal, and it could eliminate the electrochemical cross talk by the use of electrode arrays and single-enzyme label.”

“Therefore, if different antibodies were immobilized on the electrode at the same time, the low detection limit was got without enzymatic reaction; it could also avoide the use of the throwaway electrode arrays, which made the detection cheaper.”

“It is seen that the combination of the traditional electrode and a magnetic electrode could be a new strategy to generate and amplify the detecting signals for sensing.”